# The Role of Tryptophan Dysmetabolism and Quinolinic Acid in Depressive and Neurodegenerative Diseases

**DOI:** 10.3390/biom12070998

**Published:** 2022-07-18

**Authors:** Knut Hestad, Jan Alexander, Helge Rootwelt, Jan O. Aaseth

**Affiliations:** 1Department of Research, Innlandet Hospital Trust, P.O. Box 104, NO-2381 Brumunddal, Norway; jaol-aas@online.no; 2Faculty of Health and Social Sciences, Inland Norway University of Applied Sciences, P.O. Box 400, NO-2418 Elverum, Norway; 3Department of Air Pollution and Noise, Division of Climate and Environmental Health, Norwegian Institute of Public Health, P.O. Box 222, NO-0213 Oslo, Norway; jan.alexander@fhi.no; 4Department of Medical Biochemistry, Oslo University Hospital, P.O. Box 4950, NO-0424 Oslo, Norway; hrootwel@ous-hf.no

**Keywords:** tryptophan, quinolinic acid, neurodegeneration, NMDA receptor, depression, Alzheimer, Parkinson, AIDS, antioxidants, glutathione

## Abstract

Emerging evidence suggests that neuroinflammation is involved in both depression and neurodegenerative diseases. The kynurenine pathway, generating metabolites which may play a role in pathogenesis, is one of several competing pathways of tryptophan metabolism. The present article is a narrative review of tryptophan metabolism, neuroinflammation, depression, and neurodegeneration. A disturbed tryptophan metabolism with increased activity of the kynurenine pathway and production of quinolinic acid may result in deficiencies in tryptophan and derived neurotransmitters. Quinolinic acid is an N-methyl-D-aspartate receptor agonist, and raised levels in CSF, together with increased levels of inflammatory cytokines, have been reported in mood disorders. Increased quinolinic acid has also been observed in neurodegenerative diseases, including Parkinson’s disease, Alzheimer’s disease, amyotrophic lateral sclerosis, and HIV-related cognitive decline. Oxidative stress in connection with increased indole-dioxygenase (IDO) activity and kynurenine formation may contribute to inflammatory responses and the production of cytokines. Increased formation of quinolinic acid may occur at the expense of kynurenic acid and neuroprotective picolinic acid. While awaiting ongoing research on potential pharmacological interventions on tryptophan metabolism, adequate protein intake with appropriate amounts of tryptophan and antioxidants may offer protection against oxidative stress and provide a balanced set of physiological receptor ligands.

## 1. Introduction

Of the eight essential amino acids, tryptophan has the lowest reserves and is therefore more prone to deficiency during malnutrition or catabolic states than the other seven [1]. In addition to being a building block in protein biosynthesis, of which as little as 1% of ingested tryptophan is normally used, most of this aromatic substance is metabolised via several pathways to bioactive metabolites of clinical importance [2]. The best known of these pathways is enzymatic conversion to serotonin, of which >90% is normally formed in the distal part of the small intestine and the rest in the central nervous system (CNS). This pathway involves two enzyme complexes. The first and rate-limiting step is the conversion of tryptophan to 5-OH-tryptophan, catalysed by the tryptophan hydroxylase, which is then followed by the amino acid decarboxylation by aromatic-L-amino acid decarboxylase (AADC) to produce serotonin (Figure 1). Tryptophan hydroxylase catalyses the monooxygenation of tryptophan, using one oxygen atom from O_2_; the other oxygen atom is accepted by tetrahydrobiopterin. The reaction requires Fe(II). An alternative pathway is direct decarboxylation to tryptamine by intestinal bacteria or in neurons that express only AADC, and not tryptophan hydroxylase [3,4]. Tryptamine belongs to a group of endogenous “trace amines” produced by the decarboxylation of amino acids that have a high turnover rate. They act on the trace amine-associated receptors (TAARs)—a group of G-protein-coupled receptors [3]. TAAR1 is expressed in the brain, and agonists appear to modulate the classical monoaminergic neurotransmission. In recent years, attention has been increasingly focused on the medical significance of the enzymatic oxidation of tryptophan to form various indole compounds and toxic and protective metabolites within the so-called kynurenine pathway [5,6]. The kynurenine pathway leads to the production of kynurenine, and further to quinolinic acid, niacin, and nicotinamide adenine dinucleotide (NAD). The kynurenine pathway occurs mainly in the liver, its initial steps being catalysed by the tryptophan-2,3 dioxygenase (TDO), but also in the brain by the enzymes indoleamine dioxygenase 1 and 2 (IDO1, IDO2), which are more widely expressed [4]. A rate-limiting step in the kynurenine pathway catalysed by IDO uses molecular oxygen in the oxidation of tryptophan and as cofactor iron in its Fe(II) form [7]. Reactive oxygen species (ROS) may arise from the enzymatic activity of IDO [8]. Yet another pathway that occurs in the intestine is the conversion of tryptophan by the bacterial enzyme tryptophanase to indole and indole derivatives, such as indole-3-acetic acid [4].

The enzymes tryptophan dioxygenase (TDO) and indoleamine dioxygenase (IDO) use dioxygen for oxidation and Fe(II) iron as cofactor in the conversion of tryptophan to kynurenine. IDO expression is increased by TNF-alpha or INF-gamma. The activated Fe(II) forms of IDO may generate reactive oxygen species (ROS). The metabolite quinolinic acid, further down the kynurenine pathway, may act as an agonist of the NMDA receptor, whereas kynurenic acid works as an antagonist. Formation of the neuroprotective picolinic acid is an alternative pathway to quinolinic acid. Other pathways include the formation of serotonin and tryptamine; tryptamine agonises the trace amine associated receptor 1 (TAAR1) that modulates the classical monoaminergic neurotransmission and the indole pathway. Kynurenine, indole derivatives, and 6-formylindolo(3,2-b) carbazole (FICZ) bind to the aryl hydrocarbon receptor (AhR). FICZ may be formed from indole derivatives or upon exposure of tryptophan to H_2_O_2_, and it binds with an extremely high affinity to AhR (see text).

Increased flux down the kynurenine pathway with an accompanying deficiency of the substrate, tryptophan, may contribute to various pathological conditions, such as depression and anxiety [9,10,11]. In addition, some metabolites, such as quinolinic acid in the kynurenine pathway, have neurotoxic properties.

In this narrative review, we discuss tryptophan metabolites that can trigger neuroinflammation [12,13], thereby accentuating mood disorders and contributing to the development of neurodegenerative conditions. Effects of the metabolites on neuroinflammation and links to psychiatric and neurodegenerative disorders were based on a literature search, in addition to our own research. The presumed pathogenic role of the metabolite quinolinic acid is of particular interest.

## 2. Tryptophan and Indolamine Dioxygenase

Indoleamine 2,3-dioxygenase belongs to the family of oxidoreductases, which also includes tryptophan 2,3-dioxygenase (TDO), sometimes referred to as tryptophan oxygenase. Both TDO and IDO contain a non-covalently bound Fe(II) heme unit per monomer. TDO is usually tetrameric, whereas IDO is monomeric. Several researchers [14,15] have discussed the mechanisms of tryptophan oxidation. It is assumed that it involves catalytic activity of the Fe(II) heme moiety of the enzymes and that TDO and IDO react by similar mechanisms. Because the interaction of tryptophan with oxygen appears to include the formation of an intermediary ferryl compound that may act as a strong oxidant [16], the IDO reaction may contribute to oxidative stress [17,18]. ROS may lead to increased production of inflammatory cytokines (TNF-alpha and IL-6, i.e., from macrophages), further upregulating IDO and thereby instigating a vicious circle [19]. The harmful effects of ROS formed from the IDO reaction may be counteracted by such endogenous antioxidants as the glutathione (GSH)/GSH-peroxidase system and presumably also by such exogenous antioxidants as polyphenols.

Increased production of peroxide and ROS resulting from IDO activation may conceivably disrupt the finely tuned interaction between intracellular oxidants and antioxidants, concurrent with a change in the tryptophan/kynurenine ratio. Overexpression of IDO, leading to an accelerated tryptophan–kynurenine pathway and ROS production, may cause a depressed intracellular concentration of reduced GSH and disrupt the intracellular redox balance [20]. Upon interaction with ROS, intracellular GSH is rapidly oxidised to its disulfide form GSSG [21], which can either be reduced again enzymatically to GSH intracellularly or expelled from the cells [22,23], where it may form S-glutathionylated plasma proteins [24] or lead to increased extracellular levels of its amino acid constituents [25], which can be detected in blood [26]. Under the redox conditions existing in circulating blood, not only oxidised GSH, but also cysteine and homocysteine, will exist predominantly as mixed disulfides with albumin or other plasma proteins [27,28]. Kynurenine and its breakdown products, such as quinolinic acid, have diverse biological impacts, including immunological and psychological effects [11,29,30]. In addition, the interferon (IFN-α) treatment for hepatitis C may precipitate depressive symptoms, due to activation of the kynurenine pathway [31,32]. Of relevance also is the fact that kynurenine by the enzyme kynurenine-3-hydroxylase can be converted to 3-hydroxy-kynurenine, which, in addition to being a redox reactive metabolite, appears to act as a neurotoxin through its ability to induce protein modifications [33]. In addition, activation of the kynurenine pathway appears to enhance the development of cardiovascular disease [34,35].

Until recent years, medical interest in tryptophan has focused mainly on its role in serotonin synthesis. In the CNS, serotonin modulates mood and cognition. Several neuropsychiatric problems have been attributed to changes in the availability of serotonin. The focus has currently changed, however, to other aspects of tryptophan metabolism such as the indole and kynurenine pathways. There is a delicate balance between different tryptophan pathways. It is now realised that only about 1–2% of tryptophan intake is metabolised into serotonin and approximately 95% is metabolised via the kynurenine pathway. Because the two metabolic pathways of serotonin and kynurenine compete for the substrate, tryptophan, increased metabolism of tryptophan via the kynurenine pathway may result in a relative deficiency of serotonin in the CNS. Thus, people vulnerable to stress may benefit from tryptophan supplements [36].

Notably, immune activation accompanied by increased production of INFγ will increase both TDO and IDO activity and thereby channel the tryptophan metabolism into the kynurenine pathway. IDO oxidation stimulated by immune activation is apparently of significance for a variety of disorders, including neuropsychological impairments and diseases with vascular pathology [11,35].

## 3. Quinolinic Acid

Quinolinic acid [37], also known as pyridine-2,3-dicarboxylic acid, is a dicarboxylic acid with a pyridine backbone—a downstream product of the kynurenine pathway (Figure 1 and Figure 2). It is known that quinolinic acid can act as an *N*-methyl-d-aspartate (NMDA) receptor agonist [38,39].

Through its interaction with the NMDA receptor, quinolinic acid has potent neurotoxic effects. Studies have demonstrated that quinolinic acid may be involved in neurodegenerative processes in the brain, in addition to playing a role in mood disorders [40,41].

Quinolinic acid is produced in the brain by immune activation of microglia and macrophages, which contain the IDO enzyme that initiates the kynurenine pathway [18,42,43,44,45]. According to Schwarcz et al. (2012), elevated quinolinic acid levels could also lead to axonal degeneration. Its neuroactive properties have been ascribed to its NMDA receptor agonism [46]. It also acts as a pro-inflammatory mediator and as a pro-oxidative molecule [47].

Quinolinic acid formed in the CNS is unable to escape through the blood–brain barrier (BBB). In contrast, the precursor of quinolinic acid, kynurenic acid (produced primarily in the liver), and tryptophan do pass, and can increase the CNS production of quinolinic acid. The ability of quinolinic acid to stimulate NMDA receptors may lead to excitatory neurotoxicity [46]. In response to inflammation in the brain with the activation of microglia and macrophages, quinolinic acid may reach neurotoxic levels, which can lead to neuronal dysfunction. In addition to its NMDA receptor agonism, quinolinic acid can induce lipoperoxidation and cytoskeletal destabilisation [48,49]. Its toxicity specifically affects neurons located in the hippocampus, striatum, and neocortex, due to the selectivity of quinolinic acid toward NMDA receptors residing in these locations [50].

When neuroinflammation flares up, quinolinic acid is produced in excessive amounts through the activated kynurenine pathway, leading to over-excitation of the NMDA receptors, secondarily leading to neuronal influx of Ca^2+^. High levels of Ca^2+^ in neurons trigger an activation of apoptotic pathways and cell death [50].

Through its interaction with Fe(II) (Figure 3), with the formation of Fe(II)-quinolinate complexes (i.e., 1:1, 1:2, and 1:3 complexes), a subsequent interaction with oxygen and H_2_O_2_ can induce the formation of ROS, including the highly toxic hydroxyl radical ^•^OH [49,51]. This free radical causes significant oxidative stress, leading to increased glutamate release, and resulting in DNA damage and lipid peroxidation.

Quinolinic acid has also been noted to increase phosphorylation of proteins involved in cellular structures, leading to destabilisation of the cytoskeleton. The various neurocognitive disorders attributed to or aggravated by increased levels of quinolinic acid are discussed in the following paragraphs [49].

### 3.1. Mood Disorders: Depression

In post-mortem brains of patients who had suffered from major depression, the levels of quinolinic acid in the prefrontal cortex have been found to be greater than in post-mortem brains of patients without depression [53]. Because NMDA receptor antagonists possess antidepressant properties, it is tempting to propose that increased levels of quinolinic acid in patients with depression may serve a causal or aggravating role in the mood disorder through the activation of NMDA receptors. Following INFα therapy, researchers have found increased concentrations of quinolinic acid in the cerebrospinal fluid (CSF) and have noted that the concentrations correlate with the severity of depressive symptoms [53]. In addition, increased levels of quinolinic acid could play a role in impairment of the glial–neuronal network, which could be associated with the recurrent and chronic nature of bipolar depression [54]. Based on results of a meta-analysis, Marx and co-workers (2021) have suggested that there is a shift in the tryptophan metabolism from serotonin to the kynurenine pathway in psychiatric diseases and that patients with mood disorders are characterised by a higher quinolinic-acid/kynurenic-acid ratio than people without mood disorders [55]. Moreover, Brundin and co-workers (2016) found a higher quinolinic-acid/picolinic-acid ratio in the CSF of 137 patients who were exhibiting suicidal behaviour, as compared with the values in 71 healthy controls [56]. They ascribed their observation to a reduction of an alternative kynurenine pathway metabolite, picolinic acid, that counteracts the neurotoxic action of quinolinic acid [57]. The reduced concentrations of the protective metabolite, picolinic acid, in CSF were sustained over two years after a suicide attempt in depressive patients. Although kynurenic acid, another metabolite of kynurenine, was not assessed in the latter study [57], it might have worked to counteract excessive NMDA receptor activation and exert a neuroprotective action [58,59,60,61]. An antidepressant effect of kynurenic acid is in accordance with the results from an experimental study in a rat model [62].

### 3.2. Parkinson’s Disease

The initial symptoms in Parkinson’s disease may be depression and other non-motor manifestations, including impaired non-verbal communication and expressivity [63,64,65]. Quinolinic acid is a possible trigger for depression and other non-motor symptoms. Quinolinic acid neurotoxicity is also thought to play a role in the further pathogenesis of Parkinson’s disease [66]. Thus, studies have shown that quinolinic acid or its Fe-chelate is involved in degeneration of the dopaminergic neurons in the substantia nigra (SN) of Parkinson’s disease patients [67]. SN degeneration combined with iron deposition is one of the key characteristics of this disease [68,69]. It has been observed that microglia, associated with dopaminergic cells in the SN, have produced quinolinic acid when experimental Parkinson’s disease symptoms were induced in macaque monkeys [70]. Although quinolinic acid levels appear to be associated with disease severity in human Parkinson’s patients, and the kynurenine/tryptophan ratio is higher in plasma and CSF in more severe cases, an additional role of lower levels of the alleged neuroprotective substances, kynurenic acid and picolinic acid, has also been suggested [71,72].

### 3.3. Alzheimer’s Disease

A correlation between quinolinic acid levels and progression of Alzheimer’s disease has been suggested. Studies comparing the post-mortem brains of patients with and without Alzheimer’s disease have revealed higher neuronal quinolinic-acid levels among Alzheimer’s disease patients than among the controls. Quinolinic acid has been associated with neuroinflammation [73] and has been observed to increase tau phosphorylation in vitro [74]. Immunoreactivity studies have demonstrated that quinolinic acid immunoreactivity is strongest in glial cells located close to amyloid plaques and also that there is immunological cross-reactivity with neurofibrillary tangles within cortical neurons [74]. Some indole metabolites, such as indole-3-propionic acid, appear to have protective effects against amyloid-β-related toxicity [75]. Busse et al. (2018) have suggested that peripheral monocytes, which were shown to have increased quinolinic acid levels in Alzheimer’s patients, could enter the brain and contribute to excitotoxicity [76].

In a mouse model of Alzheimer’s disease, Wu et al., (2013) showed, by use of immuno-histochemistry, that the density of TDO immuno-positive cells was significantly higher in these mice than in control mice [40]. The production of quinolinic acid strongly increased in the hippocampus in a progressive and age-dependent manner in the Alzheimer’s model. Significantly higher TDO and indoleamine 2,3 dioxygenase 1 immunoreactivity was observed in the hippocampus of Alzheimer’s mice. Furthermore, TDO co-localisation with quinolinic acid, neurofibrillary tangles, and amyloid deposits was shown in the hippocampus. Their results indicate that the kynurenine pathway is over-activated in the mouse model of Alzheimer’s disease. Apparently, TDO is highly expressed in the brains of humans with Alzheimer’s disease, suggesting that TDO-mediated activation of the kynurenine pathway could be involved in the formation of neurofibrillary tangles and plaque build-up.

### 3.4. Amyotrophic Lateral Sclerosis (ALS)

It has been hypothesised that quinolinic acid contributes to the development of amyotrophic lateral sclerosis (ALS) [77]. Increased activity in the glutaminergic neurotransmission induced by the NMDA receptor appears to be involved in the pathogenesis of ALS [78]. Elevated levels of quinolinic acid have been found in the CSF and spinal cord of ALS patients [54], whereas a concomitant decrease of neuroprotective kynurenine metabolites has been reported [77].

Mood disorder with depression is an early sign of ALS; it usually precedes other symptoms and may be related to the neurotoxicity of quinolinic acid [54]. Furthermore, overstimulation of NMDA receptors by quinolinic acid has an impact on motor neurons. It has been shown in rats that quinolinic acid causes depolarisation of spinal motor neurons by interacting with the NMDA receptors [79] and that gallic acid has been found to exert a protective function [80]. It has also been reported that the microglia-mediated inflammatory pathway contributes to quinolinic-acid-induced toxicity [81]. Furthermore, quinolinic acid plays a role in mitochondrial dysfunction in motor neurons [41], and together, these effects could contribute to ALS symptoms [54]. Further research is highly recommended on a possible role of quinolinic acid and other kynurenine metabolites in the pathogenesis in both sporadic and familial forms of ALS.

### 3.5. Human Immunodeficiency Virus (HIV) and Acquired Immunodeficiency Syndrome (AIDS)

Even after the introduction of effective antiviral therapy, about 34–44% of HIV-infected patients suffer from neurocognitive disorders [82,83].

It has been observed that increased quinolinic acid levels in AIDS correlates with cognitive and motor dysfunctions. When patients were treated with zidovudine, quinolinic acid levels decreased, after which neurological improvement was noted [44].

More recent studies have found a correlation between levels of quinolinic acid in CSF and the severity of HIV-associated neurocognitive disease (HAND). Concentrations of quinolinic acid in the CSF are associated with different stages of HAND. In more severe stages of HIV, increased concentrations of quinolinic acid in the CSF of HAND patients correlates with the severity of HIV encephalitis [84]. In their review article, Kandanearatchi and Brew (2012) concluded that quinolinic acid could explain each pathogenic step of HIV-related cognitive disorder, even though other factors might have contributed [85]. 

Levels of quinolinic acid in the CSF of patients suffering from AIDS dementia can be up to twenty times higher than normal. Neurons exposed to such high levels of quinolinic acid for long periods can develop cytoskeletal abnormalities followed by cell death [86].

## 4. Tryptophan Metabolites and the Aryl Hydrocarbon Receptor (AhR)

The aryl hydrocarbon receptor (AhR), a ligand-activated transcription factor, has been preserved in the phylogenesis of vertebrates for millions of years. It was initially identified as a receptor for 2,3,7,8-tetrachloro-p-dioxin (TCDD) and related halogenated compounds and polycyclic aryl hydrocarbons (PAHs). Its primary significance was believed to be the inactivation of environmental toxic agents by stimulating the expression of cytochrome P (CYP) and other drug-metabolizing enzymes [87,88]. Recent research, however, has identified endogenously occurring compounds as ligands. One of these compounds is 6-formylindolo[3,2-b]carbazole (FICZ), which binds to AhR with the highest affinity yet reported. FICZ is formed upon exposure of tryptophane to H_2_O_2_ (Figure 1 and Figure 2) but may also form from microbial tryptophan indole derivatives [88]. Notably, sulfoconjugates of phenolic derivatives of FICZ have been observed in human urine [89]; its presence in human tissues is therefore likely but has yet to be determined. Although much less potent, other tryptophan indole metabolites, including indole-3-pyrovat and kynurenine, may act as AhR agonists. In contrast to persistent environmental toxicants, the signalling by endogenous AhR agonists is temporary, as they are oxidised by AhR-induced catabolizing enzymes [88]. Although a few endogenous compounds may antagonise AhR signalling [90,91], most endogenous AhR ligands will act as receptor agonists [87].

Activated AhR regulates a variety of vital functions, including immune functions, and it also seems to play a vital role in the brain–gut axis [88,92,93]. AhR activation of lymphoid cells affects immune responses by inducing cytokine production. A variety of studies have shown that AhR ligands regulate various T cells, antibody-producing B cells, and mast cells. In the CNS, the AhR is expressed in microglia and appears to mediate proinflammatory and anti-inflammatory effects, depending on the availability of endogenous agonists. For example, activation with FICZ attenuated the lipopolysaccharide (LPS)-activated microglia immune responses, whereas the LPS response was reduced in AhR knock-out mice [94]. Following an acute ischemic stroke in mice, neural cell AhR activation increased astrogliosis and suppressed neurogenesis, and conditional knock-out of AhR reduced these effects [95].

A diurnal variation in CYP1A1 activity and FICZ level related to food intake and intestinal microbiota appear to occur with a variation in AhR activation [88]. Dysfunction in AhR-related regulation of the immune system may result in autoimmunity and immune activation [96], and secondarily in a vicious circle, which further activates the tryptophan–kynurenine pathway.

## 5. Discussion

It is apparent from this review that dysregulation in the tryptophan metabolism is a common finding in cases of diseases with neurological involvement, mood disorders, and age-related cognitive decline. It has been suggested that quinolinic acid may promote interferon gamma (IFN γ) induced inflammation, and that kynurenine metabolites influence ageing and age-associated neurodegenerative conditions [31,97]. It is well known that gut microbial tryptophan metabolism may have an impact on brain function [4]. The apparent dysregulation of tryptophan metabolism in several disorders have triggered an interest in bioactive tryptophan metabolites and their receptors as targets of pharmacological intervention [98,99]. Manipulation of the kynurenine pathway away from neurotoxic quinolinic acid and towards the neuroprotective NMDA antagonist, kynurenic acid, is a possible therapeutic focus [100]. Nicotinylalanine has been shown to be an inhibitor of kynurenine hydroxylase, which results in decreased production of quinolinic acid, but its protective effect is questionable [101]. Therapeutic efforts have also focused on antioxidants, which have been shown to provide protection against the pro-oxidant properties of quinolinic acid [74]. Natural phenols, such as curcumin and epigallocatechin, may reduce the neurotoxicity of quinolinic acid via anti-oxidative mechanisms [102]. One might speculate that some other nutrients or supplements, such as acetylcystein, could exert protective action by promoting regeneration of the intracellular antioxidant, glutathione (GSH) [103,104]. It is noteworthy that acetylcysteine has been shown to replenish GSH in HIV infection [105], and it appears to alleviate inflammatory-related problems and improve prognosis [106,107]. Combined supplementation with tryptophan and antioxidants may represent a promising approach [108]. If we can curb the oxidative stress with antioxidants, the favourable effects of increased levels of serotonin may render positive the overall effect of administering tryptophan as a dietary supplement. A diet high in acetylcysteine, selenium, and natural antioxidants appears to be associated with reduced risk of chronic diseases [109,110] and constitutes an insufficiently investigated possibility for reducing pain- or mood-related problems and age-associated dementia [97]. Clinical evaluation of this approach requires several years of follow-up, however, and such a preventive treatment must be initiated before severe motor or cognitive symptoms have developed—in the early stages of Parkinson’s disease [69], for example. In a recent review, Tanaka and co-workers (2022) discussed the concept of neurodevelopmental disorders that may precede such diseases as schizophrenia and Parkinson’s disease, for which early intervention with kynurenine analogues may have neuroprotective effects [5] Notably, downregulation of the peripheral kynurenine pathway, as achieved by bariatric surgery in obese subjects, was found to be accompanied by reduced peripheral inflammation, as monitored by C-reactive protein, whereas neuroinflammation was not monitored in the obese subjects [111].

## 6. Conclusions and Perspectives

Disruption of the tryptophan metabolism with increased activity of the kynurenine pathway, the production of quinolinic acid, and oxidative stress may contribute to inflammatory responses and deficiencies in tryptophan and derived signalling molecules, such as serotonin, in addition to picolinic acid and kynurenic acid. Dysregulation in the tryptophan metabolism, accompanied by oxidative stress and low-grade inflammation, including neuroinflammation, has been observed in cases of diseases with neurological involvement, mood disorders, and age-related cognitive decline. Without adequate amounts of physiological ligands, the immune imbalance may be further skewed in a disadvantageous direction. Although research on the potential of pharmacological intervention channelling tryptophan metabolism downstream towards more favourable metabolic pathways is still ongoing, adequate protein intake with appropriate amounts of tryptophan and antioxidants do appear to offer protection against oxidative stress and provide a balanced set of physiological ligands for the 5-HT receptor, TAAR1 receptor, NMDA receptor, and AhR receptor. Clinical evaluation of antioxidants and/or kynurenine analogues with neuroprotective effects will be important in the future but will require several years of follow-up, and preventive treatment must be initiated early, before severe disease manifestations have developed. One future perspective is the development of therapeutic approaches beyond the current use of traditional antidepressants for mitigating mood disorders and severe neurodegeneration.

## Figures and Tables

**Figure 1 biomolecules-12-00998-f001:**
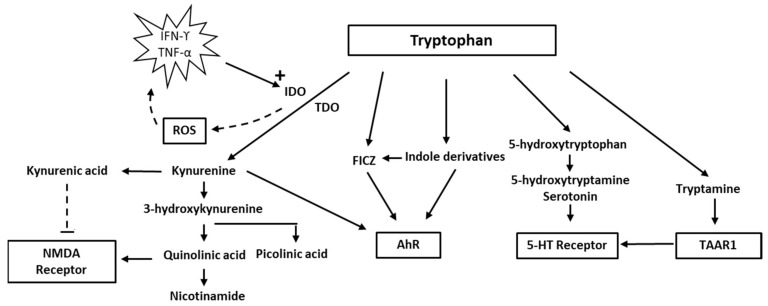
Pathways of tryptophan metabolism.

**Figure 2 biomolecules-12-00998-f002:**
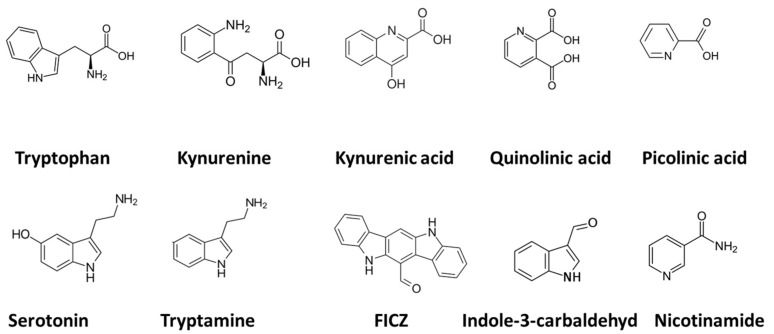
Chemical structures of tryptophan and derivatives mentioned in Figure 1. The term FICZ denotes the compound 6-formylindolo(3,2-b) carbazole.

**Figure 3 biomolecules-12-00998-f003:**
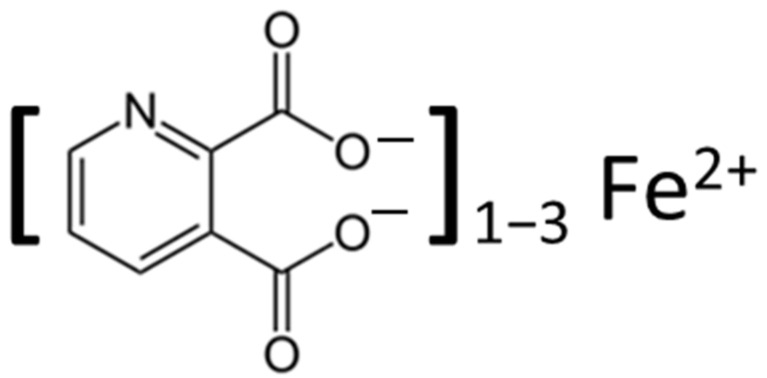
Chemical structure of the quinolinic acid iron complex. Vicinal oxygen groups (electron donors) in the molecule accounts for its coordination of Fe(II) [52], and the Fe(II)-complex can, through interaction with dioxygen and H_2_O_2_, lead to ROS generation [51,52].

## Data Availability

Not applicable.

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
