# Peer review of "The Role of Tryptophan Dysmetabolism and Quinolinic Acid in Depressive and Neurodegenerative Diseases"

_biomolecules, 2022, doi:10.3390/biom12070998_

Round 1
Reviewer 1 Report
The manuscript by Hestad and coworkers gives an overview on tryptophan metabolism and its possible consequence when the various pathways are not thoroughly regulated. I think the review is timely because indeed some of the products resulting from tryptophan modification or degradation can be neurotoxic or act as inducers of neurotoxicity. The coverage of the review is sufficiently broad and the description of the possible effects of the best known tryptophan derivatives is fair. Therefore, the manuscript will require only minor changes before recommending it for publication in IJMS, as outlined below.
1. Among the various mechanisms potentially leading to neurotoxicity one has been neglected and needs to be mentioned in the review. The tryptophan metabolite 3-hydroxykinurenine, besides being redox reactive, can be involved in pathways of protein modification similar to the neurotoxic effects induced by dysregulated dopamine and other catecholamines (see e. g. ACS Chem. Neurosci. 10, 3731, 2019).
2. Page 2, second line. Here dioxygen is included with tetrahydrobiopterin and iron(II) among cofactors of tryptophan hydroxylase. In fact, dioxygen is a substrate of the enzymatic reaction and not a cofactor of the enzyme.
3. The same mistake is repeated in the same page on the 8th line of the second paragraph, this time reporting dioxygen as a cofactor of IDO.
4. Figure 1. I suggest to include chemical formulas of the key compounds in the various metabolic pathways of tryptophan, since these will help the reader to follow the chemical reactions and further modifications described in the text.
5. Figure 1. In the legend of this figure, it is reported (last line) that FICZ binds with very high affinity but to what it binds is not mentioned.
6. Page 3, line 7. Please state ferrous or Fe(II), but not both.
7. Page 4, 5th paragraph. Here it is stated that quinolinic acid can induce lipoperoxidation and cytoskeletal destabilization; references to these studies should be given.
8. Page 4, last paragraph before Figure 2. The authors state that the interaction of quinolinic acid with Fe(II) can induce formation of ROS, but forget to mention that the reaction leading to ROS requires reaction of the Fe(II)-quinolinate complex with dioxygen.
9. In several instances throughout the paper the term tryptophan is mistyped as tryptophane (in the abstract, 6th line, in the caption of Figure 1, last line, and on p. 7, second line). Please correct. There are other minor grammar errors here and there, the Authors are suggested to check carefully the text.
Reviewer 2 Report
Tryptophan metabolism is an important issue regarding neurological and psychiatric diseases. In their review, Hestad K et al. provide an excellent overview about the dysregulation of tryptophan metabolism with its metabolites and the impact of depressive and neurodegenerative diseases.
This work has some merit because it is well-structured and easily comprehensive. The manuscript is well written and can be published in its current form.
Author Response
Review 2: Dear reviewer, thank you very much!
Reviewer 3 Report
31 May 2022
Review on the manuscript titled ‘A Role of Tryptophan Dysmetabolism and Quinolinic Acid in Depressive and Neurodegenerative Diseases’ by Hestad K et al, submitted to Biomolecules
Manuscript ID: biomolecules-1762323
Dear Authors,
Hestad and colleagues in the present review entitled ‘A Role of Tryptophan Dysmetabolism and Quinolinic Acid in Depressive and Neurodegenerative Diseases’, explored the current status of knowledge of tryptophan metabolites’ pathogenic role in the development of neurodegenerative disorders. For this purpose, they have selected some relevant evidence that focused on dysregulation in tryptophan metabolism. Findings from the collected studies showed that disruption of the tryptophan metabolism is linked to increased activity of the kynurenine pathway, production of quinolinic acid and oxidative stress, contributing to inflammatory responses, and deficiencies in tryptophan and derived signaling molecules. Finally, authors concluded by stating that pharmacological intervention channeling tryptophan metabolism downstream towards more favorable metabolic pathways is necessary to offer protection against oxidative stress.
The main strength of this manuscript is that it addresses an interesting and timely question, providing a captivating interpretation and describing how altered tryptophan metabolites can trigger neuroinflammation. In general, I think the idea of this review is really interesting and the authors’ fascinating observations on this timely topic may be of interest to the readers of Biomolecules. However, some comments, as well as some crucial evidence that should be included to support the author’s argumentation, needed to be addressed to improve the quality of the manuscript, its adequacy, and its readability prior to the publication in the present form. My overall judgment is to publish this review after the authors have carefully considered my suggestions below, in particular reshaping parts of the ‘Introduction’ and ‘Discussion’ sections by adding more evidence.
Please consider the following comments:
1. Abstract: According to the Journal’s guidelines, the abstract should be introduced as a single paragraph, and should follow the style of structured abstracts, but without headings. Please correct the actual one. The authors need to abridge it to 200 words, proportionally presenting each subsection and clearly focusing on metabolites and properties of kynurenines (KYNs) the authors intend to feature in this review. Indeed, KYN metabolites possess many and versatile biological activities. The abstract also needs to declare a type of review, narrative, scoping, or systematic. Then the following section needs to present components necessary for review type.
2. Introduction: The ‘Introduction’ section is well-written and nicely presented. Nevertheless, I believe that more detailed information about regulatory and functional aspects of the kynurenine pathway of tryptophan degradation would provide a more defined and accurate background on this topic. In this regard, it may be useful investigate kynurenine pathway alterations observed in the central nervous system as well as the periphery, its involvement in pathogenesis and disease progression (https://doi.org/10.3390/antiox11010031; https://doi.org/10.1007/s00702-022-02513-5; doi: 10.1038/s41598-020-73918-z; doi: 10.1016/j.neubiorev.2020.08.010; doi: 10.17219/acem/139572; https://doi.org/10.3390/biomedicines10040849; https://doi.org/10.1177/1545968320953671).
3. The objectives of this review are generally clear and to the point; however, I believe that there are some ambiguous points that require clarification or refining. I think that authors here need to be explicit regarding how they evaluated effect of tryptophan metabolites on neuroinflammation, and how altered tryptophan metabolism could be linked to neurodegenerative and psychiatric disorders.
4. Mood disorders: depression: I recommend presenting the most recent evidence of the involvement of KYNs in preclinical and clinical data (https://doi.org/10.1007/s43440-020-00067-5; https://doi.org/10.1038/s41380-020-00951-9).
5. Amyotrophic lateral sclerosis (ALS): I would suggest deepening information about involvement of quinolinic acid in the neuropathogenesis of amyotrophic lateral sclerosis, for example better describing how multiple biochemical phenomena associated with quinolinic acid cytotoxicity are present in ALS.
6. Discussion: In my opinion, this literature review would be more compelling and useful to a broad readership if the authors moved beyond investigating how dysregulation in tryptophan metabolism has been observed in cases of diseases with neurological involvement, mood disorders and age-related cognitive decline, and discussed theoretical and methodological avenues in need of refinement and use the evidence to suggest a path forward. In this regard, I believe that it could be useful to have more information about pathological hallmarks of neurodegenerative disorders, focusing specifically on hallmarks that underlie many PD non-motor symptoms (which often precede the motor symptoms by years or even decades). Indeed, information on the characteristics and management of non-motor symptoms (NMS) in PD (that include sensory complaints, mental disorders, sleep disturbances and autonomic dysfunction), would be necessary to truly provide a more thorough analysis on the negative impact of these complications in PD: in this regard, I suggest to add findings from studies on healthy individuals that have revealed how modulation of autonomic nervous system responses is fundamental for behavioral regulation, indicating how this function is impaired in PD patients and also affecting the preparation of adaptive motor responses required for the execution of appropriate behaviors (https://doi.org/10.1007/s00221-020-05829-4; https://doi.org/10.3390/brainsci11091203).
7. I think the ‘Conclusions’ paragraph would benefit from some thoughtful as well as in-depth considerations by the authors, because as it stands, it lists down all the main findings of the research, without really stressing the theoretical significance of the study. Authors should make an effort, trying to explain the theoretical implication as well as the translational application of their research.
8. In according to the previous comment, I would ask the authors to include a ‘Limitations and future directions’ section before the end of the manuscript, in which authors can describe in detail and report all the technical issues brought to the surface.
9. Regarding the Figures: please provide an explanatory title and caption for each figure within the text.
10. The reference list is incorrect: authors should check the Journal’s guidelines again and provide the abbreviated journal name in italics, the year of publication in bold, the volume number in italics. The review article like this needs to cite at least more than 150 to present enough evidence to claim the authors’ argument.
Overall, the manuscript contains two figures and 94 references. In my opinion, the manuscript might carry important value describing how altered tryptophan metabolites can trigger neuroinflammation.
I hope that, after these careful revisions, this paper can meet the Journal’s high standards for publication. I am available for a new round of revision of this review.
I declare no conflict of interest regarding this manuscript.
Best regards,
Reviewer
Round 2
Reviewer 3 Report
1 July 2022
The 2nd Review on the manuscript titled ‘A Role of Tryptophan Dysmetabolism and Quinolinic Acid in Depressive and Neurodegenerative Diseases’ by Hestad K et al, submitted to Biomolecules
Manuscript ID: biomolecules-1762323
Dear Authors,
In this review by Hestad and colleagues entitled ‘A Role of Tryptophan Dysmetabolism and Quinolinic Acid in Depressive and Neurodegenerative Diseases’, authors aimed to describe tryptophan metabolites’ pathogenic role in the development of neurodegenerative disorders.
I appreciated the Authors' answers to the points that I raised in the first round of review, as well as their clarifications of some of my concerns. However, despite my suggestions to provide more information, by adding some crucial studies that could have allowed to enrich and complete the theoretical framework, the authors did not add all the suggested evidence. Personally, I still believe that add findings about pathological hallmarks of neurodegenerative disorders, focusing specifically on hallmarks that underlie many PD non-motor symptoms, would help deepen the subject of this manuscript (https://doi.org/10.1007/s00221-020-05829-4; https://doi.org/10.3390/brainsci11091203; https://doi.org/10.3390/biomedicines10030627).
I hope this time authors would carefully consider my suggestions.
I am always available for other reviews of such interesting and important articles.
Thank you for your work,
Reviewer
Author Response
Reviewer #3 Round 2
"Dear Authors,
In this review by Hestad and colleagues entitled ‘A Role of Tryptophan Dysmetabolism and Quinolinic Acid in Depressive and Neurodegenerative Diseases’, authors aimed to describe tryptophan metabolites’ pathogenic role in the development of neurodegenerative disorders.
I appreciated the Authors' answers to the points that I raised in the first round of review, as well as their clarifications of some of my concerns. However, despite my suggestions to provide more information, by adding some crucial studies that could have allowed to enrich and complete the theoretical framework, the authors did not add all the suggested evidence. Personally, I still believe that add findings about pathological hallmarks of neurodegenerative disorders, focusing specifically on hallmarks that underlie many PD non-motor symptoms, would help deepen the subject of this manuscript (https://doi.org/10.1007/s00221-020-05829-4; https://doi.org/10.3390/brainsci11091203; https://doi.org/10.3390/biomedicines10030627).
I hope this time authors would carefully consider my suggestions.
I am always available for other reviews of such interesting and important articles.
Thank you for your work,"
Our Answer to Reviewer 3
Thank you for the comments.
The first paragraph of section 3b Parkinson’s disease is now changed as follows (with the recommended references inccluded):
The initial symptom in Parkinson's disease may be depression and other non-motor manifestations including impaired non-verbal communication and expressivity (Wootton, A., Starkey, N. J., & Barber, C. C. (2019). Unmoving and unmoved: experiences and consequences of impaired non-verbal expressivity in Parkinson’s patients and their spouses. Disability and Rehabilitation, 41(21), 2516-2527;
Ellena, G., Battaglia, S., & Làdavas, E. (2020). The spatial effect of fearful faces in the autonomic response. Experimental Brain Research, 238(9), 2009-2018; Borgomaneri, S., Vitale, F., Battaglia, S., & Avenanti, A. (2021). Early right motor cortex response to happy and fearful facial expressions: A tms motor-evoked potential study. Brain Sciences, 11(9), 1203.
Battaglia, S., Fabius, J. H., Moravkova, K., Fracasso, A., & Borgomaneri, S. (2022). The neurobiological correlates of gaze perception in healthy individuals and neurologic patients. Biomedicines, 10(3), 627).
Quinolinic acid is a possible trigger for depression and other non-motor symptoms.
With the reference system used in Biomolecules:
"The initial symptom in Parkinson's disease may be depression and other non-motor manifestations including impaired non-verbal communication and expressivity [63-65]. Quinolinic acid is a possible trigger for depression and other non-motor symptoms. Quinolinic acid neurotoxicity is also thought to play a role in the further pathogenesis of Parkinson’s disease [66]."